# Intravital imaging of islet Ca$^{2+}$ dynamics reveals enhanced β cell connectivity after bariatric surgery in mice

Elina Akalestou[1], Kinga Suba[1], Livia Lopez-Noriega[1], Eleni Georgiadou [1], Pauline Chabosseau[1], Alasdair Gallie[2], Asger Wretlind [3], Cristina Legido-Quigley[3], Isabelle Leclerc[1], Victoria Salem [1,4 ✉] & Guy A. Rutter [1,5,6 ✉]

Bariatric surgery improves both insulin sensitivity and secretion and can induce diabetes remission. However, the mechanisms and time courses of these changes, particularly the impact on β cell function, are difficult to monitor directly. In this study, we investigated the effect of Vertical Sleeve Gastrectomy (VSG) on β cell function in vivo by imaging Ca$^{2+}$ dynamics in islets engrafted into the anterior eye chamber. Mirroring its clinical utility, VSG in mice results in significantly improved glucose tolerance, and enhanced insulin secretion. We reveal that these benefits are underpinned by augmented β cell function and coordinated activity across the islet. These effects involve changes in circulating GLP-1 levels which may act both directly and indirectly on the β cell, in the latter case through changes in body weight. Thus, bariatric surgery leads to time-dependent increases in β cell function and intra-islet connectivity which are likely to contribute to diabetes remission.

[1] Section of Cell Biology and Functional Genomics, Imperial College London, Hammersmith Hospital Campus, London, UK. [2] Central Biological Services (CBS) Hammersmith Hospital Campus, London, UK. [3] Systems Medicine, Steno Diabetes Center, Gentofte, Copenhagen, Denmark. [4] Section of Investigative Medicine, Division of Diabetes, Endocrinology and Metabolism, Department of Metabolism, Digestion and Reproduction, Imperial College London, Hammersmith Hospital Campus, London, UK. [5] Lee Kong Chian Imperial Medical School, Nanyang Technological University, Singapore, Singapore. [6] Centre de Recherches du CHUM, University of Montreal, Montreal, QC, Canada. ✉email: v.salem@imperial.ac.uk; g.rutter@imperial.ac.uk

An estimated 415 million individuals in the world are living with diabetes, or ~1 in 11 adults[1], with Type 2 diabetes (T2D) being the most prevalent form. In the United Kingdom it is predicted that by 2025 more than five million people will be diagnosed with the disease[2]. In response to this epidemic, an abundance of pharmacological and behavioural interventions have been utilised but often focus on glycaemic control rather than long-term disease resolution[3,4]. Several clinical trials have now reported that bariatric surgery, a group of gastrointestinal procedures originally developed to aid weight loss, can induce long-term remission of diabetes, with patients often achieving normoglcaemia without the need for any more glucose-lowering medications[5–7].

Numerous studies[8–12] have attempted to unravel the mechanisms by which blood glucose control is improved post-operatively. One hypothesis implicates augmented post prandial release of gastrointestinal incretin hormones, particularly glucagon-like peptide 1 (GLP-1), which enhances glucose-stimulated insulin secretion[13–15]. As well as increases in insulin secretion, preclinical and clinical data have shown that bariatric surgery also improves both hepatic and peripheral insulin sensitivity, independent of its weight loss effects[16–19]. Nonetheless, the rapid (hours–days) reversal of dysglycaemia in human subjects treated with bariatric surgery[20,21] has provided powerful evidence that an improvement of β-cell function plays an important role in its diabetes-reversing effects. However, the exact mechanisms through which surgery impacts the β-cell, including the identity of all the extra-pancreatic signals involved, and the relative importance of changes in β-cell function and mass, have remained elusive.

A critical limitation in investigating β-cell function in living humans or preclinical models is that, in the absence of robust in vivo imaging technologies[22], function must be inferred from measurements of circulating insulin or C-peptide. These approaches preclude any quantitation of changes over time, a detailed examination of function at the level of single β-cells, or the connections between them. The latter has become an important issue since we[23] and others[24] have reported that weaker intercellular connections, and the loss of highly connected cells, that can often initiate $Ca^{2+}$ waves (sometimes referred to as "hubs"), underlie the loss of insulin secretion observed in response to challenges associated with diabetes such as gluco(-lipo)toxicity[23,25,26]. Untangling these functional changes from alterations in β-cell mass in vivo is challenging, since the latter can only reliably be determined post mortem via pancreatic biopsies, and thus at a single time point.

In an effort to overcome these limitations, the present study aimed to directly and longitudinally observe the effect of vertical sleeve gastrectomy (VSG) on pancreatic β-cell function in mice, by transplanting "reporter" islets in the anterior chamber of the eye. This approach was established by Berggren and colleagues[27] and has recently been developed by ourselves[26] to assess coordinated islet behaviour in vivo. Importantly, this technique has allowed us to image $Ca^{2+}$ dynamics recursively, in the same islet, over time and with near single-cell resolution, following surgery. We show that VSG increases β-cell $Ca^{2+}$ dynamics within 8 weeks post-surgery when compared to pre-operative baseline and a sham-operated group. Moreover, we demonstrate that VSG increases the number and strength of β to β-cell connections at 10 weeks after surgery. These changes were associated with increased circulating GLP-1 levels, suggesting that enhanced incretin production contributes to postoperative improvements in β-cell performance.

## Results

**VSG improves glucose tolerance.** Our experimental protocol is summarised in Fig. 1a. In brief, mice were placed on a high-fat high-sucrose diet (HFHSD), at 8 weeks of age (week −12), 8 weeks before sham, or VSG surgery (week 0). This protocol led to fasting hyperglycaemia, indicative of β-cell decompensation and defective insulin secretion, as expected[28]. Ins1Cre:GCaMPf[fl/fl] islets were isolated from donor mice and transplanted at week −4. Baseline islet $Ca^{2+}$ dynamics were imaged at week −1.

VSG-treated mice experienced a larger decrease in body weight versus sham-operated animals, which was statistically significant until week 6 (week 6 av. Sham $42.9 \pm 4.3$ g, av. VSG $34 \pm 2.4$ g, $p < 0.05$, $n = 4$–6), (Fig. 1b). VSG significantly increased the glucose clearance rate ($p < 0.01$ at 15 and $p < 0.001$ at 30, 60 and 90 min) as assessed by oral glucose tolerance test (OGTT) at postoperative week 8 (Fig. 1c) and intraperitoneal glucose tolerance test (IPGTT) 4 and 10 weeks post-operatively ($p < 0.05$ at min 30 and $p < 0.001$ at 60 and 90 min, Fig. 1d and Fig. 1e respectively). Strikingly, in all tolerance tests performed on VSG-treated mice, glucose peaked at 15 min post-glucose injection (3 g/kg) and dropped to baseline levels within 60 min by week 8, and near baseline levels at week 10. In contrast, in sham-operated mice, glucose peaked at 30 min and did not fully recover within the first 2 h of measurement.

**VSG improves insulin secretion and sensitivity but does not increase β-cell mass.** In order to understand the marked increase in the rate of glucose clearance in mice that had undergone VSG, we measured insulin secretion in vivo as a response to an IP glucose load (3 g/kg). Insulin secretion was increased significantly in VSG versus sham mice as early as 4 weeks post-operatively (Fig. 1f), with the observed peak at 15 min almost threefold higher compared to sham mice ($p < 0.05$). VSG mice were also significantly more insulin sensitive when compared to sham mice, as assessed by intraperitoneal insulin tolerance test at week 8 (ITT, 1.5 U/kg) ($p < 0.001$, Fig. 1g). However, pancreatic β-cell mass was not increased in the VSG group relative to sham controls at week 10 post-operatively (Supplementary Figs. 1a and 2). Notably, the ratio of α to β-cell mass was significantly higher in the VSG group, yet α-cell mass was not significantly increased (Supplementary Figs. 1b, c and 2).

**VSG enhances GLP-1 secretion.** To assess whether enhanced incretin release may contribute to the glucose-lowering effect of VSG we observed during IPGTT and OGTT, we measured plasma GLP-1 levels during fasting and 15 min following an orally administered glucose load (3 g/kg) (Fig. 1i). The concentration of GLP-1 was significantly higher in the VSG group, when compared to sham, 15 min post gavage (Fig. 1h).

**β-Cell $Ca^{2+}$ dynamics are enhanced following VSG.** In order to explore changes in β-cell function after surgery, we monitored intracellular $Ca^{2+}$ changes prospectively and in the same islets by confocal imaging of the anterior eye chamber[26,27]. $Ca^{2+}$ increases, measured at ambient blood glucose concentrations in the range $12.5 \pm 0.7$ mmol/L for both VSG-treated and sham mice, which occurred at a single or multiple site across the islet but did not advance across the islet, were defined as "oscillations" and were categorised as level 1 (Supplementary Movie 2). Increases that had a defined site of origin and spread across a few cells but not the full width of the imaged plane were defined as "partial" waves (Fig. 2bi and Supplementary Movie 4) and were categorised as level 2. Those increases spreading across the whole islet were termed "waves" (Fig. 2ai, bii and Supplementary Movies 1 and 5) and were categorised as level 3. If the latter wave type was recurrent, we defined the behaviour as a "superwave" (Fig 2biii and Supplementary Movie 6) and this was categorised as level 4. As illustrated in Fig. 3a, when imaged 0, 4 and 10 weeks after surgery, islets in sham-operated animals displayed a progressive

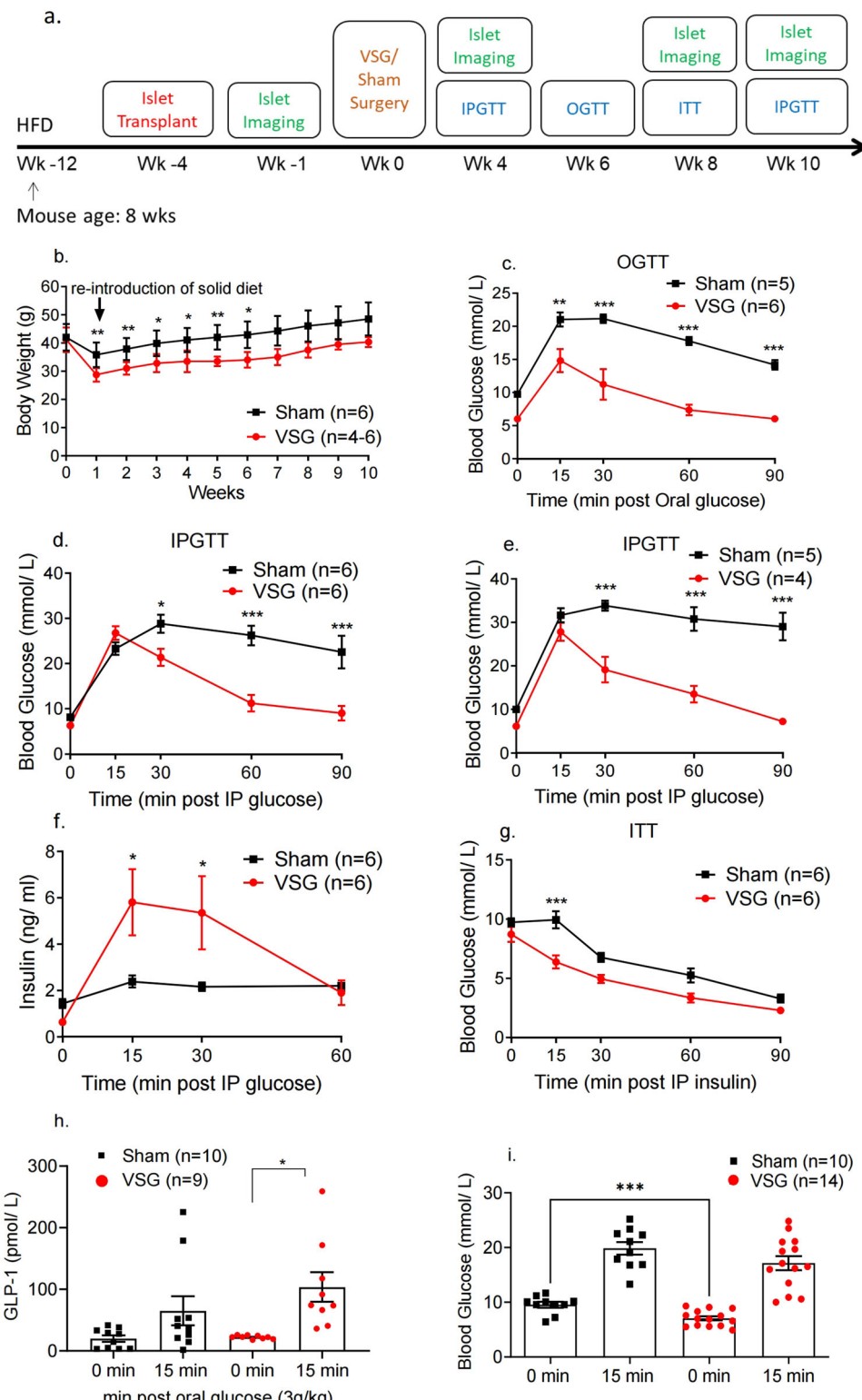

loss of coordinated $Ca^{2+}$ dynamics, as defined by the proportion of the different wave types. Thus, when imaged at 0 weeks (Fig. 2ai), wave behaviour (beginning at the bottom right; red area) area moved rapidly across the areas identified in yellow and blue. Comparable behaviour was seen at 4 weeks, with a similar site of origin of the wave (Fig. 2aii) but was lost at 10 weeks post sham surgery even though there was no significant weight difference between the two groups (Fig. 2aiii and Supplementary Movie 3).

In contrast, islets implanted into mice subject to VSG displayed sustained or gradually improving $Ca^{2+}$ dynamics following surgery. Thus, the islet shown in Fig. 2bi initially showed partial wave activity but progressed to full wave activity by week 4 (Fig. 2bii and Supplementary Movie 4) and to superwave by week 10 (Fig. 2biii and Supplementary Movie 6). Remarkably, almost all islets transplanted into VSG animals displayed either wave or superwave behaviour by week 8, even if VSG-treated animals did not display further weight loss. This is significantly higher when

**Fig. 1 VSG improves glucose and insulin tolerance in HFHSD mice. a** Timeline of procedures. **b** Body weight monitoring following VSG (red, $n = 6$ until week 6, then $n = 4$ animals) or sham surgery (black, $n = 6$ animals) (week 1: $p = 0.006$, week 2: $p = 0.008$, week 3: $p = 0.02$, week 4: $p = 0.02$, week 5: $p = 0.009$, week 6: $p = 0.03$). **c** Glucose was administered via oral gavage (3 g/kg) after mice were fasted overnight and blood glucose levels measured at 0, 15, 30, 60 and 90 min post gavage, 6 weeks after surgery, $n = 5$–6 mice/group. **d** Glucose was administered via intraperitoneal injection (3 g/kg) after mice were fasted overnight and blood glucose levels measured at 0, 15, 30, 60 and 90 min post injection, 4 weeks after surgery, $n = 6$ mice/group. $P = 0.039$ at 30 min, $p < 0.001$ for other timepoints. **e** Glucose was administered via intraperitoneal injection (3 g/kg) after mice were fasted overnight and blood glucose levels measured at 0, 15, 30, 60 and 90 min post injection, 10 weeks after surgery, $n = 4$–5 mice/group. **f** Corresponding insulin secretion levels measured on plasma samples obtained during the IPGTT performed in **d** ($n = 6$ animals), $p = 0.015$ at 15 min, $p = 0.027$ at 30 min. **g** Insulin was administered via intraperitoneal injection (1.5 UI/kg) after mice were fasted for 5 h and blood glucose levels measured at 0, 15, 30, 60 and 90 min post injection, 7–8 weeks after surgery, $n = 6$ mice/group. **h** Corresponding GLP-1 secretion levels measured on plasma samples obtained during the OGTT performed in **c** ($n = 9$–10). **i** Corresponding glucose levels for 0, 15 min obtained during the OGTT performed in **c** ($n = 10$–14 animals), $p = 0.0006$. *$P < 0.05$, **$p < 0.01$,***$p < 0.001$ VSG vs. Sham, following two-sided Student's $t$-test or two-way ANOVA (adjusted for multiple comparisons). Data are expressed as means ± SEM. VSG vertical sleeve gastrectomy, HFHSD high-fat, high-sugar diet, IPGTT intraperitoneal glucose tolerance test, ITT insulin tolerance test, OGTT oral glucose tolerance test, GLP-1 glucagon-like peptide 1. Source data are provided as a Source Data file.

compared to sham mice at the same time point ($p = 0.02$) (Fig. 3a). By week 10, the activity of all sham-transplanted islets dropped to almost zero ($p = 0.004$) (Fig. 3a). Mean wave front velocity, a measure of the speed of the wave calculated by distance (μm) divided by time (s), across the islet was not different between groups at any time point explored. Similarly, no differences were apparent between wave velocities for the different wave types in either VSG or sham operated (Fig. 3b).

**VSG maintains the number and strength of β-cell–β-cell connections.** Coordinated activity of β-cells is a feature of the healthy islet, and is likely to be important for the regulation of pulsatile insulin secretion[29]. As shown in Fig. 4a, b, Pearson-correlation analysis revealed no differences in apparent connectivity at week 0 (prior to surgery), whereas a progressive decline in connectivity was observed in the sham group. The number of connected cells (Fig. 4b) or the mean connectivity strength ($R$) (Fig. 4c, d) remained relatively constant in the VSG group, such that by week 10 these islets displayed significantly greater connectivity than the sham group (Fig. 4b). In summary, glucose-related $Ca^{2+}$ signalling in VSG mice was characterised by higher magnitude and higher sensitivity to glucose when compared with sham mice or to pre-operative levels. Conversely, the sham group which continued to gain weight and experience further deterioration in their glucose tolerance revealed a diminution in coordinated islet behaviour suggestive of glucolipotoxic effects on islet function.

**GLP-1R antagonism partly blunts the improvement in β-cell $Ca^{2+}$ dynamics after VSG.** In order to shed light on the possible mechanism(s) behind the enhanced postoperative β-cell $Ca^{2+}$ activity, we repeated the study described above, exploring the effects of a GLP-1 receptor antagonist (Fig. 5a). From week 10, all animals (VSG and sham-operated) were administered the GLP-1 receptor antagonist Exendin (9-39) (Exendin9), via a subcutaneous osmotic pump at a rate of 10 pmol/kg/min, for 2 weeks. $Ca^{2+}$ increases were monitored in the islets present in the anterior eye chamber by confocal imaging, as above, and ambient blood glucose concentrations were in the range $14.1 \pm 0.5$ mmol/L for both groups.

Following Exendin9 administration, a non-significant drop was observed in the activity (wave categories) of islets from the VSG group at week 12 (post-Exendin9) versus week 8 (pre-treatment), while in contrast a gradual increase in activity as assessed using this parameter continued in the sham group (Fig. 5b, c). A similar observation was made for the number of connected cells (Fig. 5d, e), and the mean connectivity strength ($R$) in VSG islets treated with Exendin9, when compared to week 8 (Fig. 5f–g) using baseline correction (i.e. normalisation to activity at week 0). These findings suggest that changes in circulating GLP-1 levels

may contribute to the altered islet $Ca^{2+}$ dynamics observed after VSG.

**Liragutide treatment does not fully replicate the $Ca^{2+}$ activity observed in VSG.** We sought next to test the impact of pharmacological GLP-1R agonism on islet $Ca^{2+}$ dynamics in the absence of surgery. To do so, we treated obese, diabetic mice maintained on a high-fat diet (HFD) with the pharmacological agonist liraglutide daily for 6 weeks, and used a food-restricted, weight-matched control group. Although both groups lost a similar amount of weight during the treatment period (Supplementary Fig 3a), the liraglutide group displayed significantly improved glucose tolerance versus the pair-fed group from treatment week 2 to week 6 (Supplementary Fig. 3b, c). However, islet $Ca^{2+}$ activity (Supplementary Fig. 3d), and subsequently the number of connected cells (Supplementary Fig. 3e), and the mean connectivity strength ($R$) (Supplementary Fig. 3f) were not different between the groups. In summary, additional effects of GLP-1R agonism on islet $Ca^{2+}$ dynamics, beyond those of weight loss, were not apparent.

**Mitochondrial membrane potential is not altered in ACE islets following VSG.** To further explore the mechanism(s) involved in the observed changes in islet $Ca^{2+}$ dynamics after VSG, we attempted an indirect measurement of glucose metabolism (and thus signalling) in β-cells within the engrafted islets, using the mitochondrial membrane potential sensor retramethylrhodamine methylester (TMRM). However, we did not observe any significant differences in steady-state mitochondrial membrane potential in islets from VSG versus sham-operated animals (Supplementary Fig. 2).

**Plasma lipidomic profile is altered following VSG.** We applied a targeted -omics approach to study small polar metabolite and lipid concentrations in peripheral plasma samples from sham and VSG-treated mice (Supplementary Fig. 4). Of 29 polar metabolites assessed, the levels of none were significantly altered in VSG animals when compared to sham-operated animals (Supplementary Fig. 4a).

In the lipidomic analysis, 298 lipid species from 17 different classes were studied (Supplementary Fig. 4b). Of these, one lipid from the glycerolipids class and two phosphatidylcholines, known for their role in membrane-mediated cell signalling[30], were significantly upregulated in the VSG group ($p < 0.01$).

## Discussion

Using an intravital imaging approach developed in recent years to monitor islet function in vivo[26,27], we provide here evidence that VSG causes a dramatic improvement in β-cell $Ca^{2+}$ connectivity,

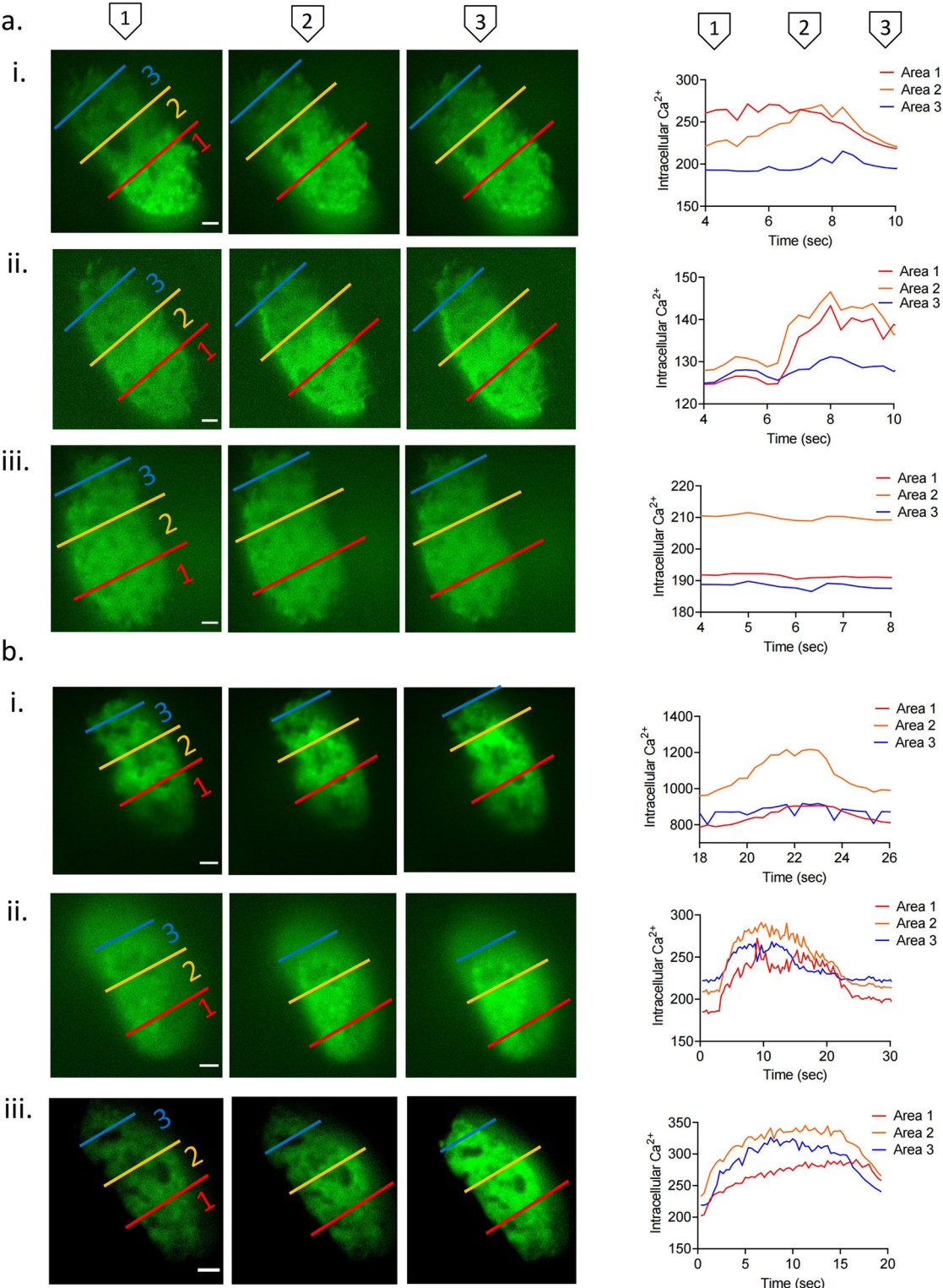

**Fig. 2 Description of Ins1Cre:GCaMPf$^{fl/fl}$ islet Ca$^{2+}$ dynamics: superwave, full wave, partial wave and no activity.** Ins1Cre:GCaMPf$^{fl/fl}$ islets implanted in the anterior chamber of the eye and imaged for 400 frames (133 s) using a spinning disk confocal microscope (see "Methods") at baseline week 0, postoperative week 4 and week 10. Each islet is separated in three regions of interest (ROIs) in order to categorise its activity. Red represents area 1 (distal islet), yellow represents area 2 (middle islet) and blue represents area 3 (proximal islet). Mean intensity is measured in each ROI for each frame (3 frames/s) and presented as Ca$^{2+}$ dynamics. **a** Ins1Cre:GCaMPf$^{fl/fl}$ islets implanted in a sham animal and imaged at (i) week 0 (full wave), (ii) 4 (full wave) and (iii) 10 (inactive). **b** (i) Ins1Cre:GCaMPf$^{fl/fl}$ islet implanted in a VSG-treated animal (i) at week 0 (partial wave), (ii) week 4 (full wave) and (iii) week 10 (superwave). Scale: 100 μm. Plasma glucose levels during imaging were 12.5 ± 0.7 mmol/L.

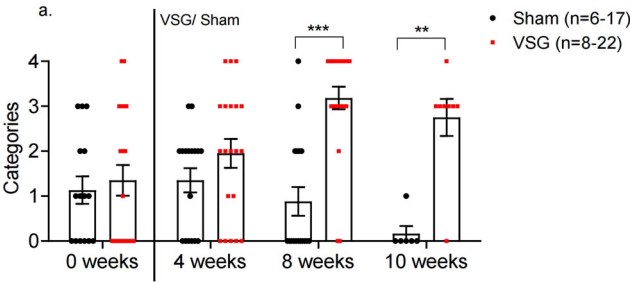

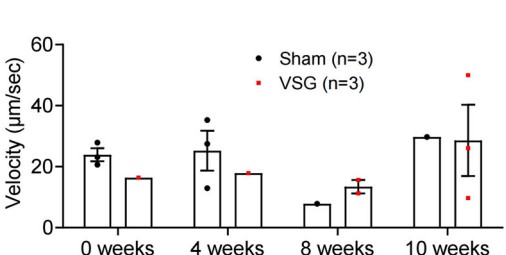

**Fig. 3 Ca²⁺ dynamics of Ins1Cre:GCaMPffl/fl islets in Sham and VSG-treated animals. a** Categorisation of Ins1Cre:GCaMPf[fl/fl] islets in sham (black, $n = 6$ animals, $n = 17$ islets) and VSG (red, $n = 10$ animals, $n = 22$ islets) up until week 8, then sham ($n = 3$ animals, $n = 6$ islets) and VSG ($n = 3$ animals, $n = 8$ islets) on week 10. Categories: 0. No activity, 1. Oscillations, 2. Partial Wave, 3. Wave, 4. Superwave. **b** Velocity of waves and partial waves of Ins1Cre:GCaMPf islets in sham and VSG animals calculated by $d/\Delta t$ and measured as μm/s sham ($n = 3$ animals, $n = 3$ islets) and VSG ($n = 3$ animals, $n = 3$ islets). **$p < 0.01$, ***$p < 0.001$ by two-sided unpaired Student's *t*-test. Data are expressed as means ± SEM. VSG vertical sleeve gastrectomy. Source data are provided as a Source Data file.

a useful assay of normal coordinated islet function and proxy for insulin secretion[22,31,32]. The use of such an approach addresses the challenges in dissecting the relative importance of the actions of bariatric surgery in changes observed in: (a) pancreatic insulin output versus peripheral insulin sensitivity, (b) β-cell function versus mass, and (c) the time courses of changes post-surgery.

Critically, we demonstrate that at similar, stimulatory glucose concentrations, islet Ca²⁺ dynamics and connectivity are dramatically increased in VSG versus sham-operated animals. Our data provide the first evidence we are aware of that alterations in β-cell function occur across the islet ensemble after surgery, and are thus likely to play a pivotal role in improving insulin output. Changes in both β-cell identity, reflecting altered gene expression[33–35], and in coordinated β-cell activity across the islet are important features of T2D[25,36]. The normalisation of either thus presents an attractive therapeutic route towards improving insulin secretion in this disease. Importantly, while several studies have demonstrated changes in islet gene expression in rodent models related to hyperglycaemia and diabetes progression, such as obese diabetic (ZDF) rats and HFHSD mice[37,38], few have examined the potential for reversing these changes as a therapeutic strategy[38,39].

Central to the present study has been the use of VSG in obese mice as a model of human bariatric surgery[40,41]. VSG is routinely offered as a treatment for human obesity, and is associated with up to 43% remission of T2D within the first postoperative year in man[42]. Our mouse model of VSG recapitulates many of the effects of bariatric surgery observed in humans, such as reduced body weight, improved glucose regulation and insulin sensitivity[43,44]. Importantly, our study had a ten-week

postoperative follow-up and, by week 8, there was no significant weight difference between the sham and VSG groups. This allowed us to separate marked improvements in insulin secretion from significant weight loss without the need to pair-feed the sham group[44]. Moreover, it corresponds with our previous findings in lean VSG-treated mice that demonstrated no weight difference when compared to sham mice at 4 weeks post-operatively, yet displayed improved glucose tolerance and corresponding insulin secretion curves during an IPGTT[45]. Of note, insulin tolerance tests at week 8 demonstrated that VSG mice had improved insulin sensitivity, an effect previously attributed in humans to rapid and significant enhancement of postoperative hepatic insulin clearance[46].

Improved glucose tolerance in VSG-treated animals was underpinned by increased insulin secretion in response to glucose at 15 and 30 min, consistent with previous studies using VSG models[40,47,48]. The fact that the insulin response to IPGTT was equally robust suggests that this effect is not solely due to an increased spike in blood glucose associated with elevated gastric emptying rates and upregulated glucose absorption, as has been previously postulated[9,49]. Increased insulin secretion in the face of lower plasma glucose demonstrates enhanced β-cell glucose sensitivity, consistent with cell-autonomous changes in islet function, alterations in circulating levels of other regulators of secretion, or an increase in β-cell number. Analysis of the endogenous pancreatic β-cell mass showed no increase in VSG versus sham-operated mice, pointing to a functional change rather than a change in endogenous β-cell mass, as underlying increased insulin output. Moreover, and though this could not be quantitated accurately due to the lack of focal distances stacking data, we saw no evidence for a change in the β-cell mass of islets engrafted into the eye. This is in line with previous findings demonstrating that there is no islet hyperplasia or increased β-cell turnover following bariatric surgery in humans or rats with obesity[50,51]. These findings contrast other studies that have reported—over similar times scales—increasing[52–54] or decreasing[48,55] β-cell mass differences, which may reflect pre-operative metabolic state or other factors.

Given many reports of increased GLP-1 release after bariatric surgery in both humans[56] and rodents[43,57], here we explored levels of both circulating fasting and post-glucose gavage GLP-1 levels. We demonstrated that GLP-1 secretion was significantly higher in the VSG group 15 min after oral glucose gavage versus sham-operated animals (Fig. 1h). Apart from increasing glucose-stimulated insulin secretion and enhancing insulin gene transcription[58,59], GLP-1 also inhibits β-cell apoptosis in animal models of diabetes[60,61]. Thus, although the underlying mechanisms remain unclear, increased GLP-1 levels might provide a partial explanation for the enhanced responses to glucose and the euglycemic effects observed after surgery. A number of studies have shown that, in postoperative patients with T2D, GLP-1 receptor (GLP-1R) blockade with the GLP-1R antagonist Exendin9 causes significant reduction of insulin secretion when compared to a control group with lower GLP-1 levels, indicating an effect on β-cell function[62,63]. Nonetheless, other studies have demonstrated that blocking GLP-1R in bariatric patients impairs glucose tolerance but not to a greater degree than before surgery, or when compared to non-operated patients[64,65]. Furthermore, Ye and colleagues[66,67] found that pharmacological or genetic blockade or elimination of GLP-1R signalling in rats or mice, respectively, had no impact on the ability of RYGB to lower body weight.

In our study, we administered Exendin9 during postoperative weeks 10–12 (when the most marked difference in islet Ca²⁺ activity between groups was observed in our first cohort) via a subcutaneous osmotic pump, in order to assess if GLP-1R antagonism will blunt

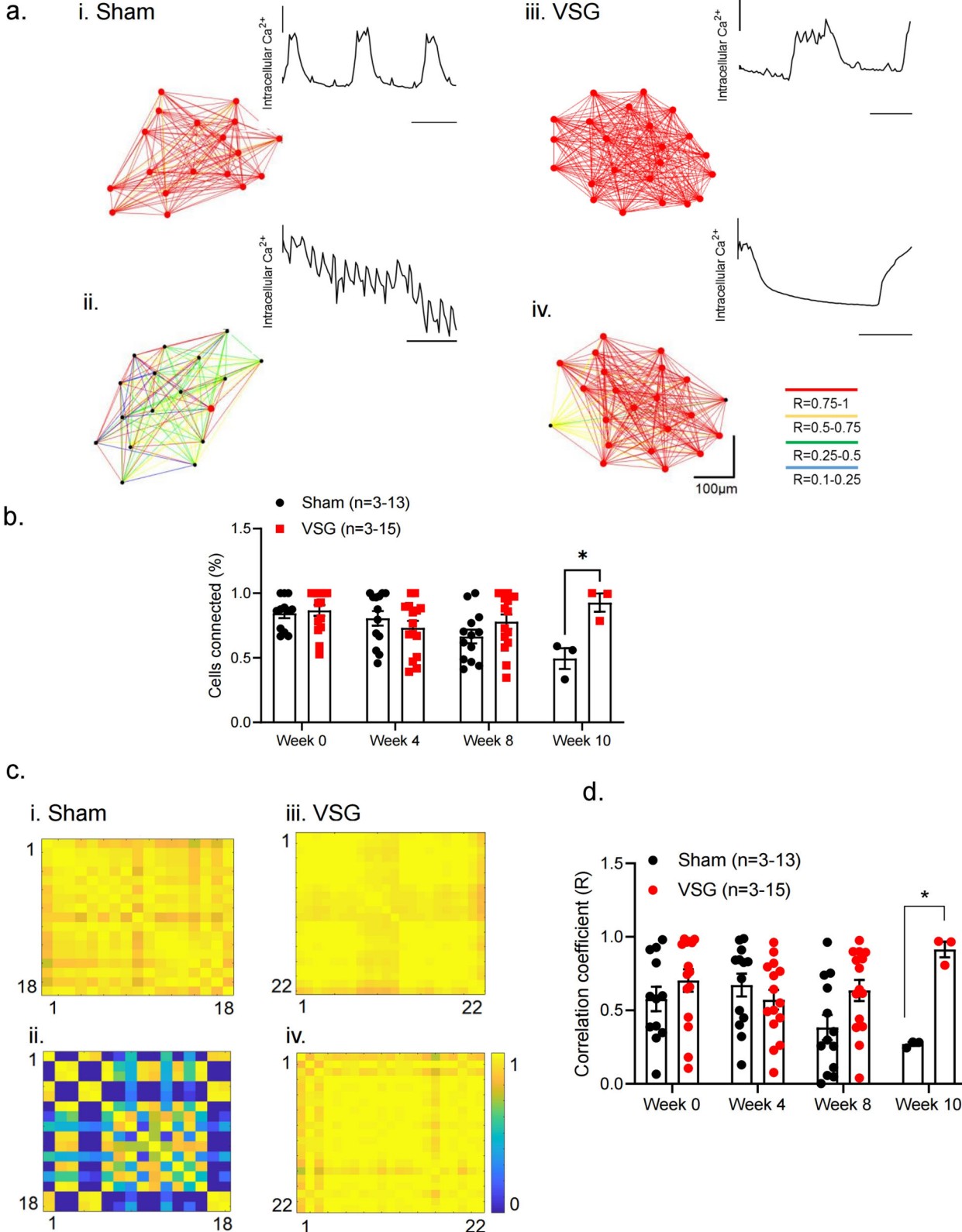

the effects on $Ca^{2+}$ signalling. In line with the view that actions of GLP-1 may be involved in the improved islet $Ca^{2+}$ dynamics, after 2 weeks of treatment Exendin9 partly reversed the differences in wave categories observed between VSG and sham groups (Fig. 5).

However, we do not exclude the possibility that a small change in body weight—which fell below significance after 8 weeks (Fig. 1b)—as well as alterations in insulin sensitivity (Fig. 1g), may also be involved in the change in β-cell function. While our -omics analysis also pointed towards an altered lipidomic profile (Supplementary Fig. 4b), the sample size was limited, restricting the conclusions that can be drawn.

Of note, differences in weight between the two groups (VSG and sham) were apparent after Exendin9 treatment (Fig. 5a). In line with this conclusion, administration of the GLP-1R agonist

**Fig. 4 Cartesian functional connectivity and correlation coefficiency of islets before and after VSG or Sham surgery. a** Cartesian functional connectivity maps displaying the correlation coefficients of β-cells within the x–y position of analysed cells (dots). (i, ii) Example of islet implanted in sham animal at week (−1) and week 10. (iii, iv) Example of islet implanted in VSG-treated animal at week (−1) and week 10. Cells are connected with a line where the strength of each cell pair correlation (the Pearson R coefficient) is colour coded: red for R of 0.76–1.0, yellow for R of 0.51–0.75, green for R of 0.26–0.5 and blue for R of 0.1–0.25. Red dots represent β-cells with the highest number of connected cell pairs. $Ca^{2+}$ activity detected during the 30 s (100 frames) imaging period analysed is displayed at the top right of each connectivity map. **b** The percentage of significantly connected cell pairs decreased significantly in the sham group at week 10 ($p = 0.02$). **c** Representative heatmaps depicting connectivity strength (Pearson R correlation) of all β-cell pairs (x–y axis) presented in (**a**). (i, ii) Example of islet implanted in sham animal (black) at week (−1) and week 10 (iii, iv). Example of islet implanted in VSG-treated animal (red) at week (−1) and week 10. R values colour coded from 0 to 1, blue to red, respectively. Red represents β-cell pairs with high connectivity strength. **d** The average of correlation coefficient (R) decreased significantly in the sham group at week 10 ($p = 0.023$). Sham group ($n = 6$ animals, $n = 13$ islets) and VSG ($n = 10$ animals, $n = 15$ islets) up until week 8, then sham ($n = 3$ animals, $n = 3$ islets) and VSG ($n = 3$ animals, $n = 3$ islets) on week 10. Data are means ± SEM and *$p < 0.05$ following two-way ANOVA (adjusted for multiple comparisons). VSG vertical sleeve gastrectomy. Source data are provided as a Source Data file.

liraglutide to HFD-fed animals which had not undergone surgery failed to fully recapitulate the alterations in islet $Ca^{2+}$ dynamics when animals were carefully weight matched by pair feeding (Supplementary Fig. 3).

An important aspect of the present study has been to examine, at the cellular level, the functional connectivity between β-cells before and after VSG or sham surgery. The percentage of significantly connected cell pairs and correlation coefficient decreased substantially in the sham group at week 10, while in the VSG group these parameters remained stable for the duration of the study. We would note that hub/follower behaviour (i.e. the existence of a "power law" in the degree of connectedness)[23] was not readily apparent in the present study. More rapid acquisition rates are likely to be needed to reveal such a hierarchy. Furthermore, we note that wave-like behaviour is more often apparent in the islet in vivo using GCaMP6f as the $Ca^{2+}$ sensor[26] than in some of our own and others' earlier studies[23,25] using entrapped, synthetic $Ca^{2+}$ probes.

Changes in metabolic signalling by glucose, particularly changes in the glycolytic and oxidative metabolism of the sugar by mitochondria, are observed in islets from human subjects with T2D[68] and are implicated in the recovery of glucose tolerance after Roux-en-Y-like gastric surgery in the mouse[69]. In the former case, this is likely to involve decreases in the expression of the critical β-cell glucose transporter, GLUT2/SLC2A2, and in the low affinity glucose phosphorylating enzyme glucokinase (GCK)[70]. Similar observations have been made in a rodent T2D model, the diabetic *db/db* mouse[71]. Measurements of gene expression were technically challenging in the current system as would have required post hoc surgical removal of islets from the eye, risking mRNA degradation and contamination with non-islet material. We instead attempted an indirect measurement of glucose metabolism, using the mitochondrial membrane potential sensor TMRM to measure this parameter in vivo. Our data with this approach did not, however, provide compelling evidence for such changes after surgery, though our sample size was relatively small. It is conceivable, therefore, that changes in the expression of genes more directly associated with the control of $Ca^{2+}$ dynamics and connectivity in and between β-cells, including Trpm2 and Gjd2 (Cx36)[69] and others associated with insulin secretion[44], play a role in the improvements observed post-surgery.

**Limitations of the study.** Although the surgical model used provides us with original information on β-cell activity via continuous monitoring, there are undoubtedly limitations in the use of islets engrafted into the ACE. These include potential differences between the vascularisation and innervation at this site compared to pancreatic in situ islets[72]. Our findings on ACE-engrafted islet reactivation following VSG are however in line

with previous results focusing on pancreatic islets isolated post mortem. Thus, Douros et al.[44] performed $Ca^{2+}$ imaging in vitro in mouse islets post mortem and showed that the percentage of islets displaying $Ca^{2+}$ oscillations in response to glucose was enhanced 2.2-fold in the VSG group, indicating increased islet glucose sensitivity after surgery. However, these earlier studies were cross-sectional in nature, and as such did not explore the apparent reactivation in vivo of individual islets and islet sub-domains, in the living animal, as described here. Finally, we note that the use of isoflurane as an anaesthetic agent has recently been linked to glucose increases[73] and might therefore have an impact on $Ca^{2+}$ measurements. In our studies, the average fed glucose was constant between VSG and sham groups at an average $13.3 ± 0.5$ mmol/L for both cohorts, and all animals were imaged at the same isoflurane dosage, yet the observed $Ca^{2+}$ dynamics were markedly different between groups.

In conclusion, our findings indicate that bariatric surgery improves glycaemic control at least partially by maintaining: (a) β-cell function and (b) coordinated activity across the islet. These effects involve changes in circulating GLP-1 levels which may act both directly and indirectly on the β-cell, in the latter case through changes in body weight. Future challenges are to understand more fully the mechanisms through which these changes are effected at the paracrine, endocrine and cellular levels.

## Methods

**Animals.** All animal procedures undertaken were approved by the British Home Office under the UK Animal (Scientific Procedures) Act 1986 (Project License PPL PA03F7F07 to I.L.) with approval from the local ethical committee (Animal Welfare and Ethics Review Board, AWERB), at the Central Biological Services (CBS) unit at the Hammersmith Campus of Imperial College London.

Adult male C57BL/6J mice (Envigo, Huntingdon, UK) were maintained under controlled temperature (21–23 °C), humidity (45–50%) and light (12:12 h light–dark schedule, lights on at 0700 hours) and were health screened clean for Federation of European Laboratory Animal Science Associations (FELASA) list pathogens. From the age of 8 weeks they were put on a 58 kcal% fat and sucrose diet (D12331, Research Diet, New Brunswick, NJ) ad libitum to induce obesity and diabetes. Four weeks after the start of this diet, the animals underwent islet transplantation into the anterior chamber of the eye of genetically modified islets expressing GCaMP6f to allow for intravital measurements of cytosolic $Ca^{2+}$. Four weeks after islet transplantation, the animals underwent either a VSG or a sham surgery as described below. The n number of the study was initially calculated using 10-week body weight (Sham: $34.5 ± 4.3$, VSG: $29.9 ± 2.5$ g) and 4-week AUC glycemia values (Sham:$1,467 ± 76$, VSG = $1,061 ± 72$ mmol/L from Garibay and Cummings[40]). This was calculated as a continuous endpoint, two independent sample study groups, α (type I error probability) 0.05 and β (type II error probability) 90%.

Ins1Cre:GCaMPf^fl/fl mice, used as donors for islet transplantation, were generated by crossing crossed Ins1Cre mice (provided by J. Ferrer, this Department of Metabolism, Digestion and Reproduction) to mice that express GCaMP6f downstream of a LoxP-flanked STOP cassette (The Jackson Laboratory, stock no. 028865). Islets donated from either sex (8 males, 6 females) were used for transplantation.

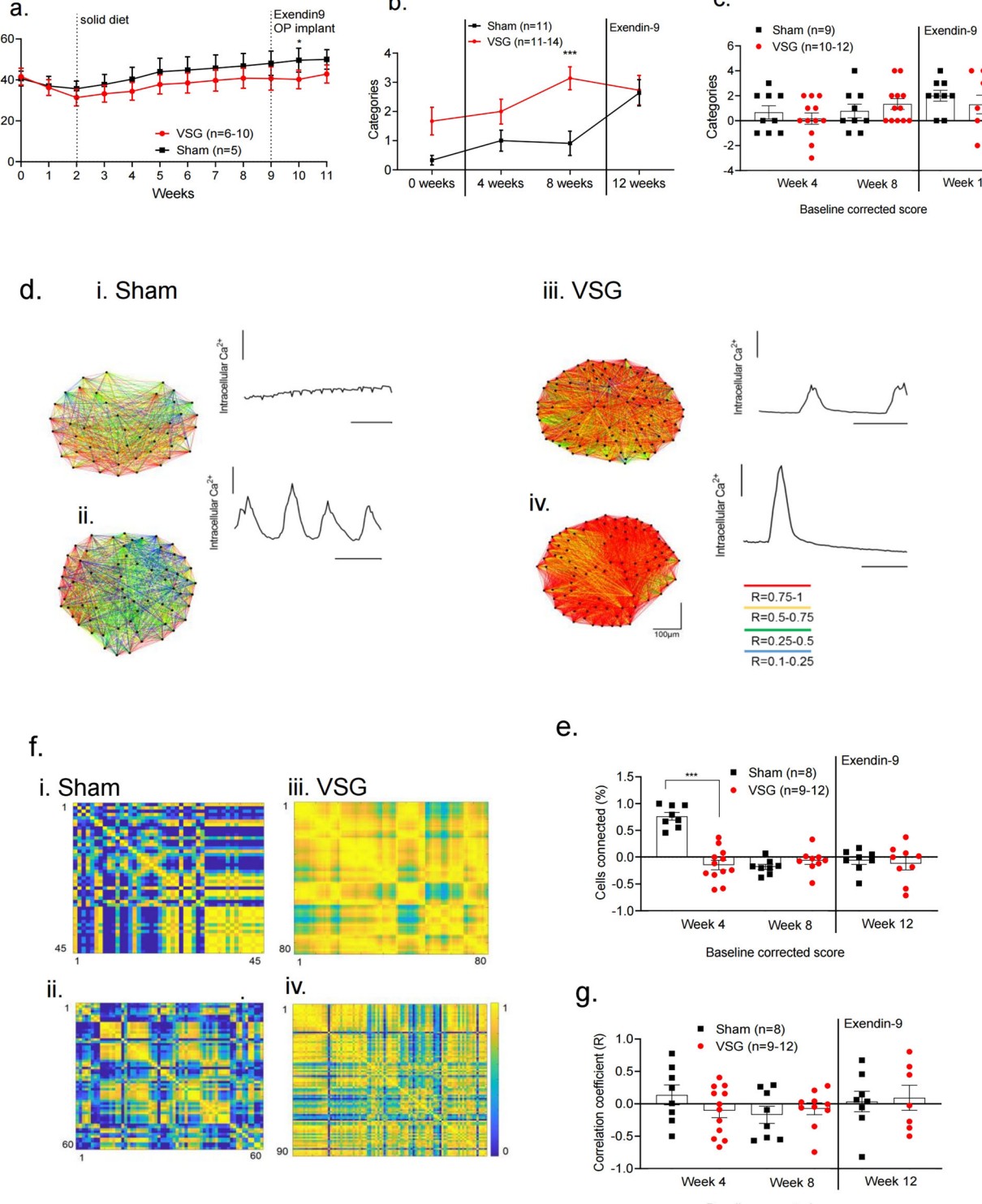

### Islet transplantation into the anterior chamber of the mouse eye (ACE).

Pancreatic islets were isolated by injecting collagenase-containing medium into the pancreatic duct, followed by separation from the exocrine tissue by gradient density and handpicking. The islets were cultured for 24 h in RPMI 1640 medium containing 11 mM glucose supplemented with 2 mM L-glutamine, 10% heat-inactivated foetal calf serum, 100 IU/mL penicillin, and 100 μg/mL streptomycin[74]. For transplantation, 10–20 islets were aspirated with a 27-gauge blunt eye cannula (BeaverVisitec, UK) connected to a 100 μL Hamilton syringe (Hamilton) via 0.4-mm polyethylene tubing (Portex Limited). Prior to surgery, mice were anaesthetised with 2–4% isoflurane (Zoetis) and placed in a stereotactic frame to stabilise the head. The cornea was incised near the junction with the sclera, being careful not to damage the iris. Then, the blunt cannula, pre-loaded with islets, was inserted into the ACE and islets were expelled (average injection volume 20 μL for 10 islets). Carprofen (Bayer, UK) and eye ointment were administered post-surgery.

### Vertical sleeve gastrectomy.

Three days before bariatric or sham surgery, animals were exposed to liquid diet (20% dextrose) and remained on this diet for up to 4 days post-operatively. Following this, mice were returned to high-fat/high-sucrose diet until euthanasia and tissues harvested 10 weeks post bariatric surgery. Anaesthesia was induced and maintained with isoflurane (1.5–2%). A laparotomy incision was made, and the stomach was isolated outside the abdominal cavity. A simple continuous pattern of suture extending through the gastric wall and along both gastric walls was placed to ensure the main blood vessels were contained.

**Fig. 5 Ca$^{2+}$ dynamics of Ins1Cre:GCaMPffl/fl islets in Sham and VSG-treated animals following Exendin9 treatment. a** Body weight monitoring following VSG (red, $n = 10$ until week 4, then 6 animals) or sham surgery (black, $n = 5$ animals). $P = 0.027$ at week 10. **b** Time course of changes in Ca$^{2+}$ activity from week 0 to week 8 ($p = 0.001$) and then following Exendin9 treatment on week 12 in sham ($n = 3$ animals, $n = 11$ islets) and VSG ($n = 7$ animals, $n = 14$ islets in weeks 4, 8 and 11 islets in weeks 0 and 12). Categories: 0. No activity, 1. Oscillations, 2. Partial Wave, 3. Wave, 4. Super Wave. **c** Baseline (week 0) categorisation of Ins1Cre:GCaMPf$^{fl/fl}$ islets in sham and VSG mice on weeks 4 and 8 and following Exendin9 treatment on week 12 (sham: $n = 3$ animals, $n = 11$ islets; VSG $n = 7$ animals, $n = 12$ islets at weeks 4 and 8 and $n = 10$ at week 12). **d** Cartesian functional connectivity maps displaying the correlation coefficients of β-cells within the $x–y$ position of analysed cells (dots). (i, ii) Example of islet implanted in sham animal at week 8 and week 12 post-Exendin9, (iii, iv) example of islet implanted in VSG-treated animal at week 8 and week 12 post-Exendin9. Cells are connected with a line where the strength of each cell pair connection ($R$) is colour coded: red for $R$ of 0.76–1.0, yellow for $R$ of 0.51–0.75, green for $R$ of 0.26–0.5 and blue for $R$ of 0.1–0.25. Red dots represent β-cells with the highest number of connected cell pairs. Ca$^{2+}$ activity detected during the 30 s (100 frames) imaging period analysed is displayed at the top right of each connectivity map. **e** Baseline (week 0) corrected percentage of significantly connected cell pairs in sham ($n = 3$ mice, 8 islets) and VSG ($n = 7$ mice, 12 islets at week 4, then 9 islets at weeks 8 and 12), respectively. **f** Representative heatmaps depicting connectivity strength of all β-cell pairs ($x–y$ axis). (i, ii) Example of islet implanted in sham animal at week 8 and week 12 post-Exendin9. (iii, iv) Example of an islet implanted in VSG-treated animal at week (8) and week 12 post-Exendin9. $R$ values are colour coded from 0 to 1, blue to red, respectively. Red represents β-cell pairs with high connectivity strength. **g** Baseline (week 0)-corrected correlation coefficient ($R$) in sham ($n = 3$ animals, $n = 8$ islets) and VSG ($n = 7$ animals, $n = 12$ islets at week 4, $n = 10$ islets at week 8 and $n = 7$ islets at week 12), respectively. Data are means ± SEM and *$p < 0.05$, ***$p < 0.001$ by two-sided Student's $t$-test. Data are expressed as means ± SEM. VSG vertical sleeve gastrectomy. Source data are provided as a Source Data file.

Approximately 60% of the stomach was removed, leaving a tubular remnant. The edges of the stomach were inverted and closed by placing two serosae only sutures, using Lembert pattern. The initial full thickness suture was subsequently removed. Sham surgeries were performed by isolating the stomach and performing a 1 mm gastrotomy on the gastric wall of the fundus. All animals received a 3-day course of analgesics Carprofen (Bayer, UK) and a 5-day course of SC antibiotic injections (enrofloxacin 10 mg/kg).

**In vivo Ca$^{2+}$ imaging of Ins1Cre:GCaMPf$^{fl/fl}$ islets in the ACE**. A minimum of 4 weeks was allowed for full implantation of islets before imaging. Imaging sessions were performed as previously described[26] with the mouse held in a stereotactic frame and the eye gently retracted, with the animal maintained under 2% isoflurane anaesthesia. All imaging experiments were conducted using a spinning disk confocal microscope (Nikon Eclipse Ti, Crest spinning disk, ×20 water dipping 1.0 NA objective). The signal from GCaMP6f fluorophore (ex. 488 nm, em. 525 ± 25 nm) was monitored in time-series experiments for up to 20 min at a rate of 3 frames/s. Ca$^{2+}$ traces were recorded for three min, with a mean blood glucose reading of $12.5 ± 0.7$ mmol/L in cohort 1 and $14.1 ± 0.5$ mmol/L in cohort 2 (Exendin9). Islets were continuously monitored, and the focus was manually adjusted to counteract movement. Animals were imaged 3 days prior to VSG (baseline) and then at 4, 8, and 10 weeks post-operatively.

**Glucose tolerance tests**. Mice were fasted overnight (total 16 h) and given free access to water. At 0900 hours, glucose (3 g/kg body weight) was administered via intraperitoneal injection or oral gavage. Blood was sampled from the tail vein at 0, 5, 15, 30, 60 and 90 min after glucose administration. Blood glucose was measured with an automatic glucometer (Accuchek; Roche, Burgess Hill, UK).

**Insulin tolerance tests**. Mice were fasted for 8 h and given free access to water. At 1500, human insulin (Actrapid, Novo Nordisk) (1.5 U/kg body weight) was administered via intraperitoneal injection. Blood was sampled from the tail vein at 0, 15, 30, 60 and 90 min after insulin administration. Blood glucose was measured with an automatic glucometer (Accuchek; Roche, Burgess Hill, UK).

**Plasma insulin and GLP-1 measurement**. To quantify circulating insulin and GLP-1 (1–37) levels, 100 μL of blood was collected from the tail vein into heparin-coated tubes (Sarstedt, Beaumont Leys, UK). Plasma was separated by sedimentation at 10,000 $g$ for 10 min (4 °C). Plasma insulin levels were measured in 5 μL aliquots and GLP-1(1–37) levels were measured in 10 μL aliquots by ELISA kits from Crystal Chem (USA).

**Osmotic pump implantation**. ALZET mini-osmotic pump Model 1002 (DURECT Corporation, Cupertino, CA, United States) were utilised for continuous subcutaneous administration. The pump had a fill volume of 100 μL and the expected pumping rate was 0.25 μL/h for 14 days. Exendin9 (Insight Biotechnology Limited, UK) was completely dissolved in 0.9% saline within 1 min and filled into the pump according to the manufacturers protocol. The final rate was approximately 10 pmol/kg/min. After filling, pumps were primed in 0.9% saline (10 mL) 37 °C for at least 3 h according to the manufacturers protocol. Each mouse received a subcutaneous implantation of one osmotic pump, placed between the shoulder blades, on week 10 following VSG or sham procedures.

**Mitochondrial membrane potential measurement**. Imaging sessions were performed as described above, with the mouse held in a stereotactic frame and the eye gently retracted, with the animal maintained under 2–4% isoflurane anaesthesia. All imaging experiments were conducted using a spinning disk confocal microscope (Nikon Eclipse Ti, Crest spinning disk, ×20 water dipping 1.0 NA objective). TMRM (ThermoFisher Scientific, MA, USA) was diluted in 0.9% saline at final concentration 0.5 mg/ kg and injected via the tail vein. The signal from GCaMP6f fluorophore (ex. 488 nm, em. 525 ± 25 nm) and TMRM (ex. 549 nm, em. 575 nm) was monitored by obtaining z-stack images of ACE islets 15 min following injection. Fluorescence quantification was achieved using ImageJ (https://imagej.nih.gov/ij/).

**Liraglutide study**. Adult male C57BL/6J mice (25–30 g) (Charles River, UK) were individually housed under controlled temperature (21–23 °C) and light (12:12 h light–dark schedule, lights on at 0700 hours) with ad libitum access to standard chow and water. Syngeneic Ins1Cre:GCaMPf$^{fl/fl}$ islets were implanted into the anterior eye chamber of their eyes. Prior to imaging experiments, animals were put on HFD for approximately 8 weeks to induce obesity and glucose intolerance. The energy density of HFD was 5.21 kcal/g and composed of 20% kcal protein, 60% kcal fat and 20% kcal carbohydrate (Research diets; D12492). A glucose intolerant phenotype was confirmed by IPGTT (1 g/kg glucose bolus), prior to baseline recordings of Ins1Cre:GCaMPf$^{fl/fl}$ islets (ex.: 488 nm; 300 ms exp. at 3 Hz). Blood glucose readings during the acquisitions averaged around 12.2(±0.9) mmol/L. Subsequently, animals were allocated to receive daily subcutaneous injections of either liraglutide (at 50nmol/kg) or vehicle (saline + Zn$^{2+}$) combined with weight-matching. Injections were administered over a period of 6 weeks. At the end of the treatment regimen, IPGTTs (1 g/kg glucose bolus) and a final recording of calcium dynamics of Ins1Cre:GCaMPf$^{fl/fl}$ islets were conducted (ex.: 488 nm; 300 ms exp. at 3 Hz). During imaging experiments, blood glucose measurements were comparable for the two groups, averaging around 10.2(±1) versus 9.2(0.2) mmol/L for liraglutide and weight-matched groups, respectively. This study was conducted under project licence PPL750462 (project licence holder Prof. Kevin Murphy).

**Immunohistochemistry of pancreas sections**. Isolated pancreata were isolated 10 weeks after surgery (age: 32 weeks) and were fixed in 10% (vol/vol) buffered formalin and embedded in paraffin wax within 24 h of removal. Slides (5 μm) were submerged sequentially in Histoclear (Sigma, UK) followed by washing in decreasing concentrations of ethanol to remove paraffin wax. Permeabilised pancreatic slices were blotted with ready-diluted anti-guinea pig insulin (Agilent Technologies, USA) and anti-mouse glucagon (Sigma, UK) primary antibody (1:1000). Slides were visualised by subsequent incubation with Alexa Fluor 488 (1:1000) and 568-labelled donkey anti-guinea pig and anti-mouse antibody (1:1000). Samples were mounted on glass slides using VectashieldTM (Vector Laboratories, USA) containing DAPI. Images were captured on a Zeiss Axio Observer.Z1 motorised inverted widefield microscope fitted with a Hamamatsu Flash 4.0 Camera using a Plan-Apochromat 206/0.8 M27 air objective with Colibri.2 LED illumination. Data acquisition was controlled with Zeiss Zen Blue 2012 Software. Fluorescence quantification was achieved using ImageJ (https://imagej.nih.gov/ij/). Whole pancreas was used to quantitate cell mass.

**Metabolic and lipidomic measurements**. Plasma was collected via cardiac puncture in all mice following a 3-day wash period post-Exendin9 administration. Molecules from plasma were measured using two approaches at Steno Diabetes Center Copenhagen. One method focused on a panel of known metabolites

previously associated with diabetes, diabetes complications and metabolic dysfunction, these molecules were fully quantified using targeted ultra-high-performance liquid-chromatography coupled triple quadrupole mass spectrometry (UHPLC-QqQ-MS/MS) as described earlier[75]. The second method aimed to measure a broad array of lipid species. Lipidomic sample preparation followed the Folch procedure with minor adjustments[76]. Lipids were measured with UHPLC coupled quadruple-time-of-flight mass spectrometry (UHPLC-QTOF/MS) in both positive and negative ionisation mode, identification was done using MZmine (version 2.28) matching to an in-house library[77], finally peak areas were normalised to internal standards[78]. Significance was tested by Student's two-tailed t-test.

**Statistical analysis**. Data were analysed using GraphPad PRISM 9.0 software. Significance was tested using non-parametric, unpaired Student's two-tailed t-tests with Bonferroni post-tests for multiple comparisons, or two-way ANOVA as indicated. $P < 0.05$ was considered significant and errors signify ±SEM.

**Pearson (R)-based connectivity and correlation analyses**. Correlation analyses in an imaged islet were performed between β-cell pairs and their extracted fluorescent $Ca^{2+}$ traces over time in MATLAB using a custom-made script. β-Cell intensity was measured in 100 consecutive frames (30 s). Regions of Interest (18–40 per islet, depending on size) were selected with single or near single-cell resolution (i.e. 10–20 μm diameter) and captured approximately 95% of the fluorescence of the image plane. A noise reduction function (effectively a rolling average) was applied to smooth noisy data and minimise the effects of outliers. The data window size used to calculate the moving averages was set to 5% of the total data points collected during each capture. The correlation coefficient $R$ between all possible (smoothed) β-cell pair combinations (excluding the autocorrelation) was assessed using Pearson's correlation. The Cartesian co-ordinates of the imaged cells were then incorporated in the construction of connectivity line maps. Cell pairs were connected with a straight line, the colour of which represented the correlation strength and was assigned to a colour-coded light–dark ramp ($R = 0.1$–0.25 [blue], 0.26–0.5 [green], $R = 0.51$–0.75 [yellow], $R = 0.76$–1.0 [red]). Cells with the highest number of possible cell pair combinations are shown in red. Data are also displayed as heatmap matrices, indicating all possible β-cell pair connections on each axis (min = 0; max = 1). The positive $R$ values (excluding the auto-correlated cells) and the percentage of cells that were connected to one another were averaged and compared between groups.

**Reporting summary**. Further information on research design is available in the Nature Research Reporting Summary linked to this article.

## Data availability
The authors declare that all data supporting the findings of this study are available within the paper and its supplementary information files. Source data are provided with this paper.

## Code availability
The custom-made script that supports the connectivity findings of this study are available in GitHub with the identifier 'Connectivity-calcium/ Pearson-correlation-coefficient-analysis'[79].

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

## Acknowledgements
The authors would like to thank Ms Bonnie Glenn and the animal technicians in Central Biological Services (CBS) unit at the Hammersmith Campus of Imperial College London for their assistance during this study. E.A. was supported by a grant from the Rosetrees Trust (M825) and from the British Society for Neuroendocrinology. G.A.R. was supported by a Wellcome Trust Investigator Award (212625/Z/18/Z), MRC Programme grants (MR/R022259/1, MR/J0003042/1, MR/L020149/1), and Experimental Challenge Grant (DIVA, MR/L02036X/1), MRC (MR/N00275X/1), and Diabetes UK (BDA/11/0004210, BDA/15/0005275, BDA 16/0005485) grants. This project has received funding from the European Union's Horizon 2020 research and innovation programme via the Innovative Medicines Initiative 2 Joint Undertaking under grant agreement No. 115881 (RHAPSODY) to G.A.R. V.S. and K.S. were supported by Harry Keen Diabetes UK Fellowship (BDA 15/0005317). I.L. was supported by a project grant from Diabetes UK (16/0005485).

## Author contributions
E.A., K.S., V.S. and L.L.-N. undertook the mouse studies. V.S. and G.A.R. designed and supervised the study. E.A. undertook all data analyses. E.G. developed connectivity scripts and P.C. contributed to connectivity analysis. A.W. and C.Q. performed the -omics analysis. A.G. and I.L assisted with mouse studies. E.A. and G.A.R. wrote the manuscript with contributions from all the authors.

## Competing interests
G.A.R. has received grant funding from Sun Pharmaceuticals Inc. and Les Laboratoires Servier and is a consultant for Sun Pharmaceuticals Inc. The remaining authors declare no competing interests.
