## [Peer Review File · Nature Communications]

Reviewer #1 (Remarks to the Author):

This manuscript takes a novel and useful approach to understanding the important impact of bariatric surgery to impact beta cell function. By implanting islets into the eyes of mice that then receive either VSG or sham surgery, they can ask critical questions about function that is independent of neuronal input and immunological insult. By in large, the data support the important impact of bariatric surgery to improve beta cell function using this unique system.

I do have a couple of concerns regarding data and data analysis:

- 1) The n's here are pretty small. This is understandable given the difficult nature of these experiments but it does present challenges to whether these data are reproducible. Power analyses for some of the key endpoints would help a reader understand how well powered these experiments actually are.
- 2) These low n's do represent a challenge to the statistical analysis. The authors use standard parametric approaches but those depend on meeting the criteria for normality. This is not presented and I would be quite surprised if they did meet normality standards. I believe it is necessary for the authors to consult a statistician and redo all of these analyses to make a stronger case that these data are reliable despite the small n's.
- 3) The authors divide the islet transplantation into three areas. It is not at all clear why this was done and whether the rationale is important. As it stands it smacks of being arbitrary and could allow for some ability to apply multiple tests to their data in a manner that may be inappropriate.
- 4) Very little information is provided on how the intensities are captured and quantified. This is not a trivial process and so more detail is required.
- 5) Imagen is done under isoflurane. Do we know whether iso can impact islet activity in the AC chamber? Iso certainly can have impacts on insulin secretion in vivo.
- 6) The authors use a suture approach to creating the VSG stomach. These approaches are known for having leaks at times. Did the authors observe the stomachs at sacrifice and was there any evidence of leaks?
- 7) This approach has a number of advantages that the authors do not discuss. The islets in the AC are presumably not innervated nor do they have much influence from the immune system. The implication is that VSG effects are independent of these factors. This conclusion would be a useful contribution to the literature.

Reviewer #2 (Remarks to the Author):

The manuscript "Intravital imaging of islet Ca²⁺ dynamics reveals enhanced beta cell connectivity after bariatric surgery in mice", NCOMMS-20-19605, suggests that Vertical Sleeve Gastrectomy (VSG) leads to time-dependent increases in beta cell function and intra-islet connectivity, together driving diabetes remission, in a weight-loss independent fashion. In vivo Ca²⁺ measurements in islets transplanted to the anterior chamber of the eye were used as readout of beta cell function. VSG improved coordinated Ca²⁺ activity in parallel with improved glucose tolerance, circulating GLP-1 and insulin secretion.

This is an interesting piece of work that however needs some more mechanistic input.

With regard to possible explanations for the observed euglycemic effect of VSG the authors

measured plasma GLP-1. They find an increase in the VSG group. However, in the discussion they state that the enhanced insulinotropic effect of GLP-1 is unlikely to explain the dramatic increase in insulin secretion observed. The main reason should be that the peak GLP-1 following oral gavage did not differ between sham and VSG mice. They also argue that enhanced insulin secretion was observed in the VSG group during IPGTT, when there is no stimulation of GLP-1 secretion. I am not sure, however, that the quality of data allows such conclusions. Few experiments and great error bars. If more convincing experiments are done allowing such conclusions to be drawn the authors need to provide experimentally supported alternative mechanisms.

In terms of using Ca²⁺ as readout for beta cell function and thereby a proxy for insulin secretion under the present experimental conditions, using isoflurane as an anesthetic, is questionable based on recent findings using a similar experimental in vivo set up (FASEB J 2020,34:945-959). If the latter findings are correct, isoflurane should have profound effects on glucose homeostasis and thereby Ca²⁺ measurements. This publication needs to be considered in light of the present findings.

In terms of the enhanced beta cell Ca²⁺ dynamics following VSG we also need some more insight. The authors conclude that glucose-related Ca²⁺ signaling in VSG mice is characterized by higher magnitude and higher sensitivity to glucose and that this might be explained by changes in glucose metabolism. I am curious to understand on what grounds the authors suggest that VSG causes changes in glucose metabolism in the islets. This should be possible to estimate by for example measuring in vivo mitochondrial membrane potential. Based on the discussion regarding insulin release, I guess that the authors also exclude GLP-1 as the responsible factor for the effects on Ca²⁺. Again, this could be tested in vivo by blocking the GLP-1 receptor. Other potential mechanisms worthwhile investigating are changes in innervation patterns and release of various transmitter substances subsequent to VSG.

Reviewer #3 (Remarks to the Author):

This manuscript provides interesting new data on the impact of bariatric surgery on in vivo beta cell function. The experiments are elegantly designed and executed. The finding that islets transplanted into the eye exhibit better beta cell connectivity and function when transplanted into VSG compared with sham is interesting and provides important new information on the islet effects of bariatric surgery. Overall, this is an excellent paper; however, there are a few limitations that should be clarified.

1. The authors need to enhance discussion of the potential mechanisms by which VSG improves beta cell function of islets located in the eye. Further, the authors need to discuss what the physiologic relevance of this finding is, given that this approach to assessing the impact of VSG on islet function is not assessing islets in their typical pancreatic location. What is the benefit of the present approach over isolating islets from sham vs VSG-operated mice and transplanting those islets into the ACE for in vivo imaging?
2. Enhancement of postprandial GLP-1 secretion is consistently found after VSG in human patients. The absence of an effect of VSG on postprandial GLP-1 is thus a limitation of the present model which should be acknowledged.
3. The authors should exercise caution in the speculation that the data are showing body weight independent effects as groups did significantly differ in body weight for most of the study.
4. A few minor points – The authors indicate that both males and females were used as islet donors; however, it is difficult to find how many of each sex are presented throughout the data. Please

indicate the number of M/F donors per group. The body weight graph indicates that the first data point is at week 1 post-op, but the matched body weights at this time point would suggest that this is at baseline (week 0)?

Reviewer #1 (Remarks to the Author):

This manuscript takes a novel and useful approach to understanding the important impact of bariatric surgery to impact beta cell function. By implanting islets into the eyes of mice that then receive either VSG or sham surgery, they can ask critical questions about function that is independent of neuronal input and immunological insult. By in large, the data support the important impact of bariatric surgery to improve beta cell function using this unique system.

>We thank the reviewer for his/her time in reviewing our manuscript, and for the helpful suggestions provided.

I do have a couple of concerns regarding data and data analysis:

1) The n's here are pretty small. This is understandable given the difficult nature of these experiments but it does present challenges to whether these data are reproducible. **Power analyses** for some of the key endpoints would help a reader understand how well powered these experiments actually are.

>The reviewer raises a fair point, which we have addressed in our revised manuscript. The n numbers required for our initial study were calculated using 10-week body weight (Sham: 34.5±4.3, VSG: 29.9±2.5g) and 4 week AUC glycemia values (Sham:1,467 ± 76, VSG = 1,061 ± 72 mmol/L from Garibay and Cummings, 2017¹). This was calculated as a continuous endpoint, 2 independent sample study groups, α (type I error probability) 0.05 and β (type II error probability) 90%. Our corresponding BW (Sham: 42.9±4.3, VSG: 34±2.5g) and 4 week AUC glycemia values (Sham: 2669±193.1, VSG: 1406±137.1 mmol/L) confirmed the sample size of n=4-6 as adequate. Our values are slightly higher than the reported values, most likely due to the different diet we used (high fat/ high sucrose). This is now specified on page 16, lines 16-20.

>For the other parameters measured, we have now performed post hoc power calculations² using ClinCalc for insulin secretion, islet Ca²⁺ activity, β -cell connectivity, and correlation coefficient which indicate power of 94-100% to detect significant changes, based on the number of experiments performed². Most importantly, however, we have now essentially repeated the whole study, and additionally explored the role of GLP-1 by administering Exendin9 on post-operative week 9. This has allowed us to replicate our Ca²⁺ findings up to week 8, and increased our n numbers by 5-8 per group.

2) These low n's do represent a challenge to the statistical analysis. The authors use standard parametric approaches but those depend on meeting the criteria for **normality**. This is not presented and I would be quite surprised if they did meet normality standards. I believe it is necessary for the authors to consult a statistician and redo all of these analyses to make a stronger case that these data are reliable despite the small n's.

>Our original mouse n numbers were indeed too low to meet normality standards, and this was confirmed after checking Gaussian distribution by using D'Agostino & Pearson and Anderson-Darling to calculate skewness and kurtosis. As a result, all stats comparing VSG and Sham groups presented in the manuscript were calculated using non-parametric, unpaired Student t-tests, as appropriate in these circumstances. This is now specified in the manuscript in page 21, line 6-7.

As mentioned in response to point #1 above, and in line with the reviewer's suggestion, we have now performed new experiments to give substantially increased n-values, with a total of 11 shams, 12 VSGs up to 8 weeks. Our experimental protocol after this time point then differed (to deal with the queries from Ref #2, below).

3) The authors divide the islet transplantation **into three areas**. It is not at all clear why this was done and whether the rationale is important. As it stands it smacks of being arbitrary and could allow for some ability to apply multiple tests to their data in a manner that may be inappropriate.

>Thank you for raising this important point. The approach we used in figure 2, where waves were observed across the whole islet, followed a reasonably standard procedure to segment the image into three regions of interest³ and allowed us to show wave progress conveniently between the regions defined as time series in the accompanying plots. We must emphasise that the same regions were not defined for all islets examined. Instead, for our subsequent analysis (e.g. Figs 3a, 5b) each islet was categorized according to the nature of the observed Ca²⁺ increases. If Ca²⁺ increases occurred at a single or multiple site across the islet, but did not then advance across the whole islet (or, to be precise, the optical section of the islet which was imaged), then these changes were defined as “oscillations”. Increases that had a defined site of origin but did not spread across the full width of the imaged plane, were defined as “partial” waves. Those increases spreading across the whole islet were termed “waves”. If the latter wave type was recurrent, we defined the behaviour as a “super wave”. These definitions are specified in page 7, line 14-21.

4) Very little information is provided **on how the intensities are captured and quantified**. This is not a trivial process and so more detail is required.

>More information on imaging procedure and FIJI quantification are now provided in materials and methods on page 18, lines 1-5 and on page 21, line 12-17

5) Imagine is done under **isofluorane**. Do we know whether iso can impact islet activity in the AC chamber? Iso certainly can have impacts on insulin secretion in vivo.

>We have now expanded on the use of isoflurane in our “Study Limitations” section on page 15, lines 14-19.

6) The authors use a suture approach to creating the VSG stomach. These approaches are known for having **leaks** at times. Did the authors observe the stomachs at sacrifice and was there any evidence of leaks?

>Thank you for raising this important point. In the UK it is unfortunately not possible legally to use a staples gun for preclinical models: we are only allowed the sutures approach. Indeed, micro-leaks can occur but are usually resolved during healing and treatment with antibiotics. If not, the animal is euthanised within the first three days. In our technique, we report the use of Lembert double suture for the first time, a pattern that allows inverted and dense sealing of the edges of the stomach. This technique limits both the danger of leak and of infection, as bacteria found on the stomach edges do not come into contact with the abdominal cavity organs. In both our lean studies (euthanised 4 weeks post-op) and our HFD studies (euthanised 10/12 weeks post-op) the stomachs observed post mortem showed no evidence of a leak, potentially due to the fast healing as made possible by suture material, size and pattern.

7) This approach has a number of advantages that the authors do not discuss. The islets in the AC are presumably not innervated nor do they have much influence from **the immune system**. The implication is that VSG effects are independent of these factors. This conclusion would be a useful contribution to the literature.

>We are grateful to the referee for this observation, and indeed he/she is quite right that this is a criticism sometimes levelled at the ACE engraftment approach. However, islets in the eye *do* become

both efficiently vascularised and innervated⁴, though of course the identity of the neurones is likely to differ from those that innervate the pancreas. We have now modified the Discussion to make the excellent point that changes in neural input may contribute to the improvements in beta cell function post-VSG.

Reviewer #2

The manuscript “Intravital imaging of islet Ca²⁺ dynamics reveals enhanced beta cell connectivity after bariatric surgery in mice”, NCOMMS-20-19605, suggests that Vertical Sleeve Gastrectomy (VSG) leads to time-dependent increases in beta cell function and intra-islet connectivity, together driving diabetes remission, in a weight-loss independent fashion. In vivo Ca²⁺ measurements in islets transplanted to the anterior chamber of the eye were used as readout of beta cell function. VSG improved coordinated Ca²⁺ activity in parallel with improved glucose tolerance, circulating GLP-1 and insulin secretion.

This is an interesting piece of work that however needs some more mechanistic input.

>We are grateful to the reviewer for taking the time to carefully review our manuscript, and for the helpful suggestions provided.

With regard to possible explanations for the observed euglycemic effect of VSG the authors measured plasma GLP-1. They find an increase in the VSG group. However, in the discussion they state that the enhanced insulinotropic effect of GLP-1 is unlikely to explain the dramatic increase in insulin secretion observed. The main reason should be that the peak GLP-1 following oral gavage did not differ between sham and VSG mice. They also argue that enhanced insulin secretion was observed in the VSG group during IPGTT, when there is no stimulation of GLP-1 secretion. I am not sure, however, that the quality of data allows such conclusions. **Few experiments and great error bars. If more convincing experiments are done allowing such conclusions** to be drawn the authors need to provide experimentally supported alternative mechanisms.

>We thank the reviewer for raising this very important point. In response, we have now performed further experiments and indeed are able now to report a significant increase in GLP-1 in the VSG versus the sham group. We suspect that our ability to detect this difference in the new experiments is due both to having a larger sample size and to the inclusion of aprotinin in the protocol.

In terms of using Ca²⁺ as readout for beta cell function and thereby a proxy for insulin secretion under the present experimental conditions, using **isoflurane** as an anesthetic, is questionable based on recent findings using a similar experimental in vivo set up (FASEB J 2020,34:945-959). If the latter findings are correct, isoflurane should have profound effects on glucose homeostasis and thereby Ca²⁺ measurements. This publication needs to be considered in light of the present findings.

>The reviewer again makes a very important point. We have now cited the publication above in our “Study Limitations” section on page 15, lines 14-19. Importantly, the cited study shows that isoflurane does cause a glucose increase but not fluctuations in insulin secretion in anaesthetized mice.

Specifically, the study shows that isoflurane-anesthetized mice exhibited a noticeable increase in fasting basal blood glucose level (comparison between time: - 15 minutes and 0) prior to glucose stimulation. The basal fasting blood glucose reported appears to rise from ~8mmol/ L (-15min) to ~12mmol/L (0min) which is roughly equal to the average fed glucose we recorded across our imaging sessions (12.5±0.7. mmol/L). Moreover, the max time of our imaging sessions was 20 min, depending

on the number of islets implanted in each mouse, with most of our mice kept under anaesthesia for approximately 10 min. Our own findings are therefore in line with the data reported but predict that isoflurane-induced glucose fluctuations would not be dramatic enough to alter our Ca²⁺ recordings given the short time-frame and lack of an externally administered glucose stimulus. Finally, it is worth mentioning that both our groups (VSG and sham) were imaged under the same conditions in terms of isoflurane dosage, and average glucose – whilst the difference between the Ca²⁺ dynamics in each case remains highly significant.

In terms of the enhanced beta cell Ca²⁺ dynamics following VSG we also need some more insight. The authors conclude that glucose-related Ca²⁺ signalling in VSG mice is characterized by higher magnitude and higher sensitivity to glucose and that this might be explained by changes in glucose metabolism. I am curious to understand on what grounds the authors suggest that VSG causes changes in glucose metabolism in the islets. This should be possible to estimate by for example measuring **in vivo mitochondrial membrane potential**.

>We thank the reviewer for this very useful suggestion, and are happy to provide further clarification of our rationale, and new data. In particular, we have now inserted into the Discussion (page 14, lines 17-27) references to relevant literature, noting that decreases in the expression of genes associated with beta cell glucose sensing and intracellular metabolism, such as *Glut2/Slc2a2*, are observed in type 2 diabetes in man⁵ and in multiple rodent models of the disease⁶. Such changes in human islets are also associated with compromised oxidative metabolism of glucose, when this is measured using radiotracers⁵. Conversely, a recent study has reported a correlation between plasma glucose levels and islet expression of *Glut2* in an alternative model of bariatric surgery⁷.

In our latest cohort, we therefore attempted an indirect measurement of glucose metabolism by beta cells, using the mitochondrial membrane potential sensor TMRM to measure this parameter *in vivo*. This approach did not, however, provide compelling evidence for any such changes in the VSG model, though our sample size was relatively small. The results and discussion of this experiment are given in detail in page 14, lines 21-27. Moreover, we applied a targeted -omics approach to study small polar metabolite and lipid concentrations in plasma samples obtained from sham and VSG-treated mice. Metabolite levels were not significantly altered following VSG, although we noted that one glycerolipid and two phosphatidylcholines were significantly increased. The results are now discussed on page 10, lines 132-21.

Based on the discussion regarding insulin release, I guess that the authors also exclude GLP-1 as the responsible factor for the effects on Ca²⁺. Again, this could be tested *in vivo* by **blocking the GLP-1 receptor**. Other potential mechanisms worthwhile investigating are changes in innervation patterns and release of various transmitter substances subsequent to VSG.

>We are again grateful to the reviewer for this suggestion. We have now repeated the study and administered the GLP-1 receptor antagonist Exendin9 for two weeks via an osmotic pump. Following this, we measured Ca²⁺ activity in the engrafted islets. Further details of this experiment are now discussed on pages 9, 13. Moreover, we measured metabolic and lipid differences in plasma at 12 weeks following surgery, now presented on page 10. Together we believe these findings provide important further mechanistic insight.

Reviewer #3

This manuscript provides interesting new data on the impact of bariatric surgery on *in vivo* beta cell function. The experiments are elegantly designed and executed. The finding that islets transplanted

into the eye exhibit better beta cell connectivity and function when transplanted into VSG compared with sham is interesting and provides important new information on the islet effects of bariatric surgery. Overall, this is an excellent paper; however, there are a few limitations that should be clarified.

>We are very grateful to the reviewer for this very kind and supportive summary, and for the helpful suggestions provided.

1. The authors need to **enhance discussion of the potential mechanisms** by which VSG improves beta cell function of islets located in the eye.

>We agree with the reviewer, and have now expanded our Discussion, and importantly performed more mechanistic experiments, including Exendin9 administration post-surgery and mitochondrial membrane potential measurements, as discussed on pages 9, 13. Importantly, our findings imply a significant role for changes in GLP-1 levels in mediating the effects of the surgery, though at this stage leave open the extent to which these are direct (via binding to GLP-1R on the beta cell) or indirect. Thus, after Ex9 administration from week 8, we observed the abolition of a difference in Ca^{2+} wave categories between the VSG and sham groups, with this largely stemming from a halt in the trend towards greater Ca^{2+} dynamics (i.e. in the mean “category” level for waves) – seen up to week 8 in both groups - in the VSG group. This is presented both as a time course (Fig. 5, panel b) and as an analysis of baseline-corrected values (Fig 5 panel c). Similar findings were made for connectivity, with the trend being most apparent for the count of connected cells (Fig 5 e) and less marked for correlation coefficient (Fig.5 f).

Further, the authors need to discuss what the physiologic relevance of this finding is, given that this approach to assessing the impact of VSG on islet function is not assessing islets in their typical pancreatic location.

What is the benefit of the present approach over **isolating islets from sham vs VSG-operated mice and transplanting those islets into the ACE for in vivo imaging?**

>The referee raises an important point. Indeed, the islets are not examined in their in situ pancreatic location as it is very difficult to achieve continuous monitoring in the pancreas over several weeks. Possible alternative techniques would involve either (a) islet isolation (and hence a “snap shot” from a single animal rather than the retrospective analysis we have achieved), as suggested and as performed recently by others ⁸ or (b) alternatively the surgical insertion of an abdominal window ⁹ which we feel is likely to be very difficult given the already considerable surgical challenge imposed by the bypass surgery. We would note that alternatives such as pancreas externalisation ¹⁰ would be unsatisfactory as, again, this is a terminal procedure. On balance, therefore, we feel that the approach of use the eye as the niche for engraftment and visualise represents the optimal balance of feasibility and physiological relevance.

2. Enhancement of postprandial GLP-1 secretion is consistently found after VSG in human patients. **The absence of an effect of VSG on postprandial GLP-1 is thus a limitation of the present model which should be acknowledged.**

>We thank the reviewer for raising this very important point. In response we have now performed further experiments and indeed now are able to report a significant increase in GLP-1 in the VSG versus the sham group – these differences are likely due to the inclusion now of aprotinin in the protocol as well as the impact of increasing n-numbers.

3. The authors should **exercise caution in the speculation that the data are showing body weight independent** effects as groups did significantly differ in body weight for most of the study.

>We fully agree. Indeed, although subtle, there are body weight differences in both our original cohort and the latest cohort presented in Figure 5. Nevertheless, these differences between the VSG and sham groups became non-significant over time in cohort 1 (Fig. 1b), though, importantly, were significant (at least at week 10) after Ex9 treatment (cohort 2; Fig. 5a).

The latter difference, we feel, may be of particular importance in the context of our new, and separate set of experiments, in which we introduced the pharmacological GLP-1R agonist liraglutide in high fat fed (but non surgically-operated) mice which were then fed to ensure matching weights. Although we observed, as expected, higher glucose tolerance in the GLP1-R agonist-treated group Supp Fig. 4b, c) no difference was apparent in Ca²⁺ wave behaviour or islet connectivity between the two groups. Thus, it would appear unlikely that GLP-1R agonism alone drives the improvement in glucose tolerance in the liraglutide group .

These points are acknowledged in the Discussion, page 13, line 20-27, and we have removed the statement regarding weight loss independence from the end of the Abstract.

4. A few minor points – The authors indicate that both males and females were used as islet donors; however, it is difficult to find how many of each sex are presented throughout the data.

Please **indicate the number of M/F donors per group**.

>This is now defined in page 16, line 24, although donor sex does not affect implantation success.

The body weight graph indicates that the first data point is at **week 1 post-op**, but the matched body weights at this time point would suggest that this is at baseline (week 0)?

>Thank you for pointing this out, this has now been updated to week 0.

References:

- 1 Garibay, D. & Cummings, B. P. A Murine Model of Vertical Sleeve Gastrectomy. *J Vis Exp*, doi:10.3791/56534 (2017).
- 2 Kane, S. *ClinCalc*, <<https://clincalc.com/stats/Power.aspx>.> (
- 3 Rutter, G. A. *et al.* Subcellular imaging of intramitochondrial Ca²⁺ with recombinant targeted aequorin: significance for the regulation of pyruvate dehydrogenase activity. *Proc Natl Acad Sci U S A* **93**, 5489-5494, doi:10.1073/pnas.93.11.5489 (1996).
- 4 Leibiger, I. B., Caicedo, A. & Berggren, P. O. Non-invasive in vivo imaging of pancreatic beta-cell function and survival - a perspective. *Acta Physiol (Oxf)* **204**, 178-185, doi:10.1111/j.1748-1716.2011.02301.x (2012).
- 5 Del Guerra, S. *et al.* Functional and molecular defects of pancreatic islets in human type 2 diabetes. *Diabetes* **54**, 727-735, doi:10.2337/diabetes.54.3.727 (2005).

- 6 Thorens, B., Wu, Y. J., Leahy, J. L. & Weir, G. C. The loss of GLUT2 expression by glucose-unresponsive beta cells of db/db mice is reversible and is induced by the diabetic environment. *J Clin Invest* **90**, 77-85, doi:10.1172/JCI115858 (1992).
- 7 Amouyal, C. *et al.* A surrogate of Roux-en-Y gastric bypass (the enterogastro anastomosis surgery) regulates multiple beta-cell pathways during resolution of diabetes in ob/ob mice. *EBioMedicine* **58**, 102895, doi:10.1016/j.ebiom.2020.102895 (2020).
- 8 Douros, J. D. *et al.* Sleeve gastrectomy rapidly enhances islet function independently of body weight. *JCI Insight* **4**, doi:10.1172/jci.insight.126688 (2019).
- 9 Reissaus, C. A. *et al.* A Versatile, Portable Intravital Microscopy Platform for Studying Beta-cell Biology In Vivo. *Sci Rep* **9**, 8449, doi:10.1038/s41598-019-44777-0 (2019).
- 10 Mehta, Z. B. *et al.* Remote control of glucose homeostasis in vivo using photopharmacology. *Sci Rep* **7**, 291, doi:10.1038/s41598-017-00397-0 (2017).

Reviewer #1 (Remarks to the Author):

My biggest concern about the low n has been addressed. While I don't agree with all of their conclusions, the discussion section adequately addresses the issues.

Reviewer #2 (Remarks to the Author):

I have now read the revised version of the manuscript NCOMMS-20-19605A "Intravital imaging of islet Ca²⁺ dynamics reveals enhanced beta cell connectivity after bariatric surgery in mice" by Akalestou et al.

The authors have addressed all my concerns and I have no further comments.

Reviewer #3 (Remarks to the Author):

The authors have carefully responded to all author comments and the manuscript is much improved! I have no further suggestions.

Reviewer #4 (Remarks to the Author):

The paper summarizes a combination of established techniques brought together in a novel fashion to provide a better understanding of changes in beta cell function after bariatric surgery. Specifically, calcium dynamics within transplanted pancreatic beta cells are measured in mice who have and have not had bariatric surgery. The authors should be commended for the design and execution of an experiment with the thoroughness that is shown in Figure 1. The ability to connect these data, as a time course, to the images of the calcium dynamics of the transplanted cells, and the observation of critical and significant differences between the sham and the VSG groups is the key contribution of the paper. As written, however, these contributions are dominantly observational - and attempts to dig deeper into the mechanism driving the observations - through administration of an agonist, the measurement of membrane potential, and the metabolite and lipid profiling of plasma, did not result in any increase in the understanding of the results.

The metabolite and lipid profiling appear to be particularly superfluous to the paper - since they are written as being completed without any hypothesis as to expected changes, and hence when none are observed (in the case of metabolites) it is not clear whether this is an expected or unexpected result. Although mildly statistically significant changes were observed in a few of the monitored lipids, these have a very small fold change, and no attempt was made to link these lipids to the other observations in the paper. Additionally since there is no listing of monitored lipids and metabolites it is not clear whether all membrane-mediated cell signaling lipids were altered or only a few specific ones. The use of only three samples for the lipid and metabolite studies would strongly limit the conclusions that could be made by any results positive or negative.

While I realized that some of these "negative" experiments were in response to previous reviewers' comments - I believe that the reviewers believed the experiments would help shed light on the mechanisms which resulted in the core observations. None of the results, however, were illuminating and given this, is it not clear whether resultant changes improve the paper or not, and

hence, whether it would be better to published the core experiment and its results in absence of mechanistic explanation - and complete a more comprehensive look at multiple possible explanations of mechanism separately. I do note that my expertise is in assay development and data analysis and that I would defer to a separate reviewer with expertise in biology of diabetes for the overall importance and this publication worthiness of the observations of paper without any insight in mechanism.

A few smaller points.

The definition of the “waves” could be improved by greater attempts to reduce the observation to a hard metric - exactly as is done for cell connectivity. The movies certainly are helpful but the description would be improved by attempts to reduce to a number. As it is written, while the phenomena are clear, it is not obvious all observations easily fall into each category.

In the description for Figure 1 the reporting of p-values could be improved - ideally with the use of the actual values rather than the repeated < sign with different marks. In the sentence with the 15, 30 60, and 90 weeks it is not clear why they are split up given it is the same value.

In the first sentence though seemingly trivial, going beyond US and UK for figures would be good.

RESPONSES TO REVIEWERS' COMMENTS

Reviewer #1 (Remarks to the Author):

My biggest concern about the low n has been addressed. While I don't agree with all of their conclusions, the discussion section adequately addresses the issues.

>We thank the reviewer for supporting publication of our revised manuscript

Reviewer #2 (Remarks to the Author):

I have now read the revised version of the manuscript NCOMMS-20-19605A "Intravital imaging of islet Ca²⁺ dynamics reveals enhanced beta cell connectivity after bariatric surgery in mice" by Akalestou et al.

The authors have addressed all my concerns and I have no further comments.

>We thank the reviewer for supporting publication of our revised manuscript

Reviewer #3 (Remarks to the Author):

The authors have carefully responded to all author comments and the manuscript is much improved! I have no further suggestions.

>We thank the reviewer for supporting publication of our revised manuscript

Reviewer #4 (Remarks to the Author):

The paper summarizes a combination of established techniques brought together in a novel fashion to provide a better understanding of changes in beta cell function after bariatric surgery. Specifically, calcium dynamics within transplanted pancreatic beta cells are measured in mice who have and have not had bariatric surgery. The authors should be commended for the design and execution of an experiment with the thoroughness that is shown in Figure 1. The ability to connect these data, as a time course, to the images of the calcium dynamics of the transplanted cells, and the observation of critical and significant differences between the sham and the VSG groups is the key contribution of the paper. As written, however, these contributions are dominantly observational - and attempts to dig deeper into the mechanism driving the observations - through administration of an agonist, the measurement of membrane potential, and the metabolite and lipid profiling of plasma, did not result in any increase in the understanding of the results.

The metabolite and lipid profiling appear to be particularly superfluous to the paper - since they are written as being completed without any hypothesis as to expected changes, and hence when none are observed (in the case of metabolites) it is not clear whether this is an expected or unexpected result. Although mildly statistically significant changes were observed in a few of the monitored lipids, these have a very small fold change, and no attempt was made to link these lipids to the other observations in the paper. Additionally since there is no listing of monitored lipids and metabolites it is not clear whether all membrane-mediated cell signaling lipids were altered or only a few specific

ones. The use of only three samples for the lipid and metabolite studies would strongly limit the conclusions that could be made by any results positive or negative.

While I realized that some of these “negative” experiments were in response to previous reviewers’ comments - I believe that the reviewers believed the experiments would help shed light on the mechanisms which resulted in the core observations. None of the results, however, were illuminating and given this, is it not clear whether resultant changes improve the paper or not, and hence, whether it would be better to published the core experiment and its results in absence of mechanistic explanation - and complete a more comprehensive look at multiple possible explanations of mechanism separately. I do note that my expertise is in assay development and data analysis and that I would defer to a separate reviewer with expertise in biology of diabetes for the overall importance and this publication worthiness of the observations of paper without any insight in mechanism.

>We thank the new reviewer for these comments, and we tend to agree. We have moderated our descriptions of these results accordingly, removing text on page 14, lines 5-7 and 13-16.

A few smaller points.

The definition of the “waves” could be improved by greater attempts to reduce the observation to a hard metric - exactly as is done for cell connectivity. The movies certainly are helpful but the description would be improved by attempts to reduce to a number. As it is written, while the phenomena are clear, it is not obvious all observations easily fall into each category.

>We take the reviewer’s point but would note that we have already defined wave categories (allowing numerical comparisons between conditions). However, and to further help the reader, we now extend our description of the “waving” behaviour in the Results section, assigning the categories shown in Figure 3. These clarifications are provided on page 7, lines 16-22.

In the description for Figure 1 the reporting of p-values could be improved - ideally with the use of the actual values rather than the repeated < sign with different marks. In the sentence with the 15, 30 60, and 90 weeks it is not clear why they are split up given it is the same value.

>This is now provided. P values for Figure 1 are added in legend.

In the first sentence though seemingly trivial, going beyond US and UK for figures would be good.

>International values for diabetes prevalence are now provided on page 4, lines 2-4.